# Using population-specific add-on polymorphisms to improve genotype imputation in underrepresented populations

Zhi Ming Xu[1,2], Sina Rüeger[1,2], Michaela Zwyer[3,4], Daniela Brites[3,4], Hellen Hiza[3,4,5], Miriam Reinhard[3,4], Liliana Rutaihwa[3,4], Sonia Borrell[3,4], Faima Isihaka[5], Hosiana Temba[5], Thomas Maroa[5], Rastard Naftari[5], Jerry Hella[5], Mohamed Sasamalo[5], Klaus Reither[3,4], Damien Portevin[3,4], Sebastien Gagneux[3,4], Jacques Fellay[1,2,6]*

1 School of Life Sciences, École Polytechnique Fédérale de Lausanne, Lausanne, Switzerland, 2 Swiss Institute of Bioinformatics, Lausanne, Switzerland, 3 Swiss Tropical and Public Health Institute, Basel, Switzerland, 4 University of Basel, Basel, Switzerland, 5 Ifakara Health Institute, Dar es Salaam, Tanzania, 6 Precision Medicine Unit, Lausanne University Hospital and University of Lausanne, Lausanne, Switzerland

* jacques.fellay@epfl.ch

**Data Availability Statement:** Individual level whole-genome sequencing data has been archived at the European Genome-phenome archive (EGA). Requests for data access can be made through

## Abstract

Genome-wide association studies rely on the statistical inference of untyped variants, called imputation, to increase the coverage of genotyping arrays. However, the results are often suboptimal in populations underrepresented in existing reference panels and array designs, since the selected single nucleotide polymorphisms (SNPs) may fail to capture population-specific haplotype structures, hence the full extent of common genetic variation. Here, we propose to sequence the full genomes of a small subset of an underrepresented study cohort to inform the selection of population-specific add-on tag SNPs and to generate an internal population-specific imputation reference panel, such that the remaining array-geno-typed cohort could be more accurately imputed. Using a Tanzania-based cohort as a proof-of-concept, we demonstrate the validity of our approach by showing improvements in impu-tation accuracy after the addition of our designed add-on tags to the base H3Africa array.

## Author summary

Genome-wide association studies, which study the association between genetic variants and various phenotypes, typically rely on genotyping arrays. Only a small proportion of genetic variants within the genome are typed on genotyping arrays. Untyped variants are statistically inferred through a process known as genotype imputation, where correlations between variants (haplotypes) observed in external reference panels are leveraged to infer untyped variants in the study population. However, for study populations that are under-represented in existing reference panels, the quality of imputation is often sub-optimal. This is because typed variants incorporated on existing genotyping arrays can be unsuit-able for the study population, and haplotype structures can be different between the refer-ence and the study population. Here, we illustrate an approach to select a custom set of

EGA under study accession number: EGAS00001005850 (https://ega-archive.org/studies/EGAS00001005850). Software code and a list of add-on SNPs designed for the TB-DAR cohort is available at: https://github.com/zmx21/h3africa-addon. Summary statistics are available at: https://doi.org/10.5281/zenodo.5638566.

**Funding:** This work was supported by the Swiss National Science Foundation (https://www.snf.ch; Grant No: CRSII5_177163, 310030_188888) and the European Research Council (https://erc.europa.eu/; Grant No: 883582). The funders had no role in study design, data collection and analysis, decision to publish, or preparation of the manuscript.

**Competing interests:** The authors have declared that no competing interests exist.

population-specific typed variants to improve genotype imputation in such underrepresented populations.

## Introduction

By mapping the associations between single-nucleotide polymorphisms (SNPs) and various phenotypes, genome-wide association studies (GWAS) have allowed us to gain unprecedented knowledge on the genetic basis of various human diseases and traits. An important prerequisite to conducting GWAS is the availability of a cost-effective yet accurate high-throughput genotyping method. Genotyping arrays have been used widely over the past 15 years, including in many studies facilitated by biobank resources such as the UK Biobank [1]. However, genotyping arrays rely on the imputation of a sparse set of tag SNPs (e.g. millions of SNPs) to achieve acceptable density genome-wide (e.g. tens of millions of variants). The quality of imputation is dependent on the suitability of the tag SNPs and the similarity of haplotype structure between the reference panel and the study population [2–5].

For study populations where a genetically similar reference panel or population-specific array content may not be available, whole-genome sequencing (WGS) offers an alternative to genotyping arrays. Previous studies have suggested that WGS may offer substantial gains in such a scenario, potentially pinpointing loci absent in GWAS conducted using genotyping arrays [6, 7]. However, due to the large sample sizes often required to gain sufficient statistical power in GWAS, the cost of WGS can still be prohibitive despite its recent decrease [8].

An alternative to WGS is the development of population-specific reference panels and genotyping arrays. For example, African-specific reference panels and genotyping arrays have been developed in recent years in an attempt to rectify the underrepresentation of African populations in genetic studies [9–11]. Notably, the Human Heredity and Health in Africa (H3Africa) consortium has developed the H3Africa genotyping array, which contains approximately 2.2 million tags to capture genetic variability observed in various African populations [12]. Furthermore, the African Genome Resources (AFGR) reference panel has been designed to capture the haplotype structure of various African populations to improve imputation accuracy. However, given that only a subset of African populations are represented in current publicly available reference panels, there remains African populations for which imputation is suboptimal. This is exacerbated by the fact that the level of genetic diversity is much higher among African populations compared to non-African populations, driven by the long evolutionary history and lack of bottlenecks [13, 14]. Thus, to achieve similar imputation accuracy across all African populations as to non-African populations (e.g. European or Asian), larger and more diverse imputation reference panels are needed to capture the full extent of variation [15]. For the remaining underrepresented populations, we propose the use of a combination of add-on tags and an internal population-specific reference panel as a cost-effective approach to improve genotype imputation.

For a GWAS cohort, we propose to perform WGS in a small subset (e.g. 10% of the entire cohort) for two purposes. Firstly, array manufacturers often allow researchers to add customised content to existing array designs. For example, Illumina offers the flexibility to add 5000 or 20,000 probes for the commercially available H3Africa array. In our proposed approach, the WGS data would be leveraged to determine population-specific linkage disequilibrium (LD) structures, and thus enable the selection of add-on tags to improve genotype imputation. Secondly, the strategy to supplement external reference panels with WGS samples from an internal study cohort has been employed by previous studies [16, 17]. Specifically, it has been

shown that the addition of even a relatively small number of samples from the internal cohort leads to improved imputation accuracy, especially if the study population is genetically dissimilar from the populations captured by existing reference panels [18, 19]. In our proposed approach, the WGS data would be used to construct an internal population-specific reference panel that supplements existing publicly available reference panels. This would function jointly with the selected add-on tags to further improve genotype imputation.

Given that the population that an existing genotyping array is designed for (array population) is different from the population one would like to genotype (study population), we envision that the add-on tags selected by our proposed approach would improve imputation accuracy under the following scenarios. Firstly, the minor allele of a target variant could be rare in the array population but common in the study population, and thus not captured by the existing array content. The WGS data of the study population could be leverage to select an add-on tag SNP that is in strong linkage disequilibrium (LD) with the target variant (S1(A) Fig). The internal reference panel could then be leverage to impute the target variant. Secondly, the minor allele of an existing tag could be common in the array population but rare in the study population. In such a scenario, the existing tag would be insufficient to tag the target variant, and an additional add-on tag specific to the study population would be required. (S1(B) Fig). Finally, the haplotype structure may be different between the array population and the study population. An existing tag may be on the same haplotype block as the target variant in the array population, but a different haplotype block in the study population. Thus, an add-on tag on the same haplotype block as the target variant in the study population could be added. The internal reference panel could then be leveraged to impute the target variant. (S1(C) Fig).

As a proof-of-concept example, we utilize 116 high coverage WGS samples from participants of the TB-DAR cohort (Tuberculosis patients recruited in a hospital in Dar es Salaam, Tanzania). Since the Tanzanian population has not been incorporated in existing reference panels and array designs, including the AFGR reference panel and the H3Africa genotyping array, this cohort provides an ideal basis to evaluate our approach. We first illustrate the necessity for add-on tags by calculating the genetic differentiation between our Tanzanian cohort and populations represented in the AFGR reference panel. Using the AFGR reference panel, we show that fewer sites could be successfully imputed for Tanzanian individuals compared to individuals from Sub-Saharan African populations that are represented in the reference panel. We proceed to leverage the WGS data to construct an internal reference panel that captures Tanzanian specific haplotype structures, and to select add-on tags that target common variants in the Tanzanian population that are poorly imputed under the base H3Africa array. We then confirm the validity of our approach by evaluating the improvement in imputation accuracy enabled by the addition of add-on tags, and compare our approach to add-on tags selected by random and by the Tagger software. We use both the internal Tanzanian reference panel and the external H3Africa reference panel for imputation, where for each site the genotype call is derived based on the reference panel with a higher predicted imputation accuracy for the site. Finally, we present an alternative selection scheme for mitochondrial and Y chromosome variants. We show that mitochondrial haplogroup calling can be improved, while the coverage of the Y chromosome on the H3Africa is mostly sufficient for accurate haplogroup calling. A summary of our proposed approach can be found in Fig 1.

## Results

### Performance of external reference panels

To choose the most suitable external reference panel for the TB-DAR cohort, we masked the TB-DAR WGS data to include only sites genotyped under the H3Africa array. We then

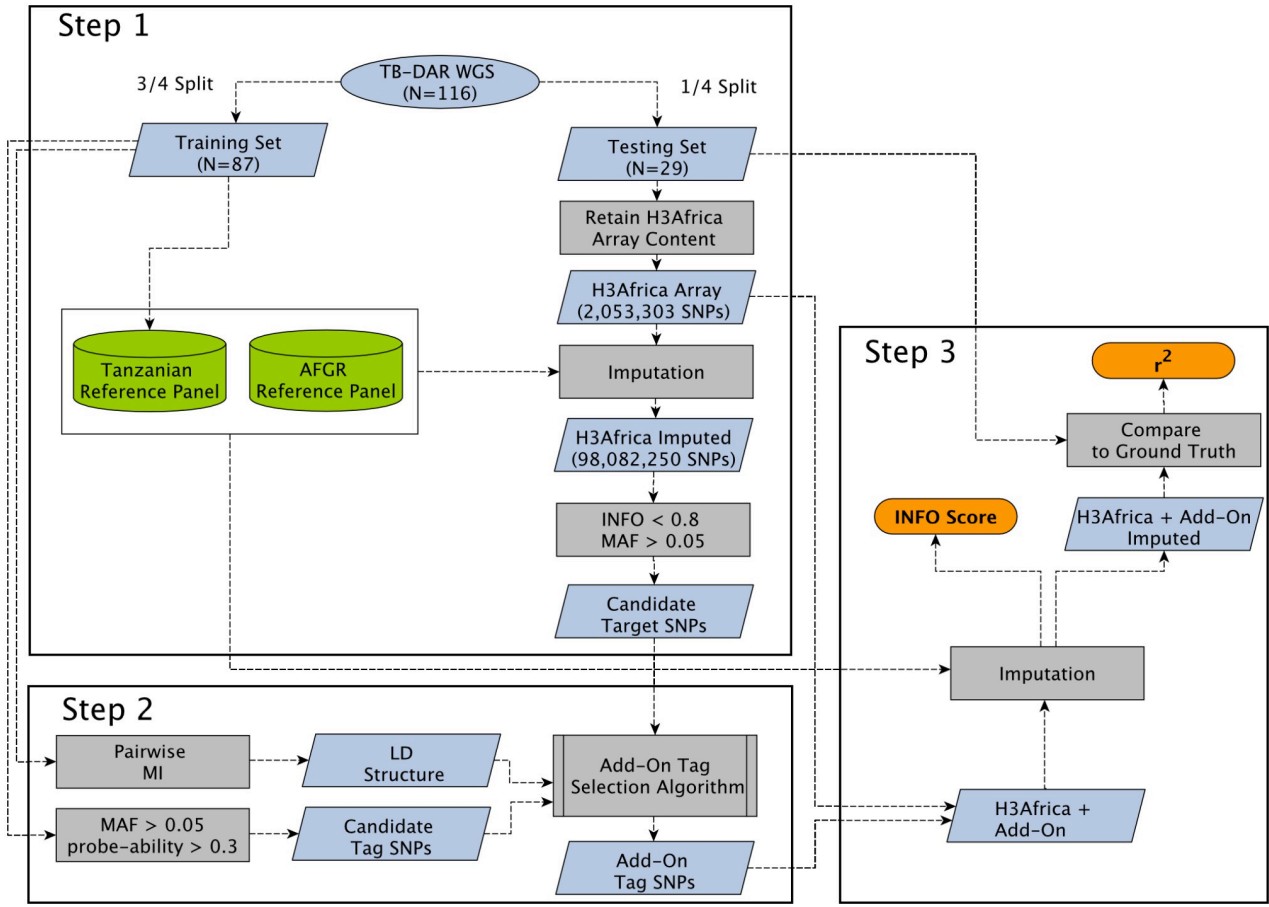

**Fig 1. Schematic of our add-on tag SNP selection procedures, with steps illustrating. Step 1)** Constructing a Tanzanian reference panel. Identifying candidate target variants, which are derived from poorly imputed variants when the H3Africa array is imputed based on the Tanzanian and AFGR reference panel. **Step 2)** Selecting add-on tags that optimally tag candidate target variants based on population-specific LD structures, allele frequencies, and probe qualities. **Step 3)** Evaluating improvements in imputation performance after adding add-on tags to the base H3Africa array. Calculating imputation quality metrics, including INFO score and $r^2$ (correlation between imputed and sequencing-based genotypes). WGS, Whole-Genome Sequencing; AFGR, African Genome Resource; MAF, Minor Allele Frequency; MI, Mutual Information; LD, Linkage Disequilibrium.

imputed the masked data and compared the imputation performance of three publicly available reference panels (as of September 2019) that include individuals of African ancestry: 1) The AFGR (African Genome Resource) reference panel hosted on the Sanger Imputation Server, which consists of 4956 individuals in total with ∼2000 individuals from Uganda, ∼100 individuals from other African populations, and the remaining from the African and Non-African populations of the 1000 Genomes project. 2) the CAAPA (Consortium on Asthma among African-ancestry Populations in the Americas) reference panel [20] hosted on the Michigan Imputation Server, which consists of 883 African American individuals, 3) the HRC (Haplotype Reference Consortium) reference panel [21] hosted on the Sanger Imputation Server, which consists of 32,470 individuals in total originating from the African and Non-African populations of the 1000 Genomes Project and other sources with predominantly individuals of European ancestry.

Table 1 illustrates the imputation performance of the three reference panels, measured by mean imputation quality (INFO Score or $r^2$), mean correlation with ground truth ($r^2$), and the fraction of variants observed in the WGS data that were successfully imputed (Imputation

**Table 1. Imputation performance of publicly available reference panels when applied to the TB-DAR data based on the H3Africa array content.** Minor allele frequency (MAF) is based on the frequency observed in the TB-DAR cohort. Imputation quality (Subcolumn 1) is measured by either INFO score (AFGR and HRC; Sanger Imputation Server) or $r^2$ (CAAPA; Michigan Imputation Server). Correlation with ground truth (Subcolumn 2) measures the correlation between the imputed dosage and the ground truth WGS dosage using the squared pearson correlation coefficient ($r^2$). Percent of variants imputed (Subcolumn 3) represents the fraction of variants observed in the TB-DAR WGS data that were successfully imputed (Imputation Quality > 0.8).

| | | AFGR | | | CAAPA | | | HRC | | |
|---|---|---|---|---|---|---|---|---|---|---|
| | | Quality (INFO) | Ground Truth $r^2$ | % Variants Imputed | Quality ($r^2$) | Ground Truth $r^2$ | % Variants Imputed | Quality (INFO) | Ground Truth $r^2$ | % Variants Imputed |
| MAF | | | | | | | | | | |
| | 0.01–0.05 | 0.95 | 0.91 | 88.6 | 0.85 | 0.83 | 67.5 | 0.88 | 0.80 | 68.3 |
| | 0.05–0.1 | 0.98 | 0.96 | 93.4 | 0.93 | 0.91 | 86.8 | 0.96 | 0.90 | 91.9 |
| | 0.1–0.5 | 0.99 | 0.97 | 92.7 | 0.96 | 0.95 | 90.1 | 0.98 | 0.95 | 91.7 |

Quality > 0.8). We observed that the AFGR reference panel outperformed both the CAAPA and HRC reference panel across MAF bins in all three metrics, and thus chose to use AFGR as the external reference panel in this study.

## Differentiation between the Tanzanian population and other African populations

Study participants of the TB-DAR WGS cohort originated from various ethnic groups within Tanzania (S1 Table). A majority of participants belonged to the Bantu-speaking ethnic groups ($N = 108$, 93.1%), with a small minority that belonged to the Nilotic ($N = 1$, 0.8%) and Cushitic ($N = 3$, 2.6%) speaking ethnic groups. Self-reported ethnic information was not available for four participants.

To measure the population differentiation between the TB-DAR WGS cohort and populations represented in the AFGR reference panel, for each pair of populations we calculated the genome-wide pairwise fixation index ($F_{ST}$). The $F_{ST}$ estimate is a metric that ranges from 0 to 1 that quantifies the degree of genetic differentiation between populations, with 0 indicating no differentiation and 1 indicating complete differentiation. Fig 2A illustrates the pairwise $F_{ST}$ measures between the TB-DAR cohort and Sub-Saharan African populations of the 1000 Genomes (1KG) project, along with the sampling locations of populations represented in the AFGR reference panel. $F_{ST}$ measures were only calculated for Sub-Saharan African 1KG populations that are also part of the AFGR reference panel, since WGS data for populations based in Uganda, Ethiopia, Namibia, and South Africa (Zulu) were not publicly available. In general, genetic differentiation was greater between populations that are further away geographically. For example, TB-DAR displayed the least differentiation with the Bantu-speaking Luhya population (LWK) in neighbouring Kenya, but the most differentiation with West African populations such as the Gambian in the Western Division of Gambia (GWD) and the Mende in Sierra Leone (MSL). A similar pattern was observed among Sub-Saharan African populations of 1KG (Fig 2B). The least differentiation was observed between population pairs in the same geographic region (e.g. YRI and ESN: $F_{ST} = 0.0008$). The most differentiation was observed between East and West African populations (e.g. LWK and GWD: $F_{ST} = 0.011$). However, since many diverse African populations were not included in the analysis, this is an underestimate of the full extent of genetic diversity within Africa. More comprehensive analyses have estimated a mean $F_{ST}$ of 0.027 between East and West African populations [22]. In addition, the genetic principal components (PCs) shown in S2 Fig also illustrate a similar pattern, where

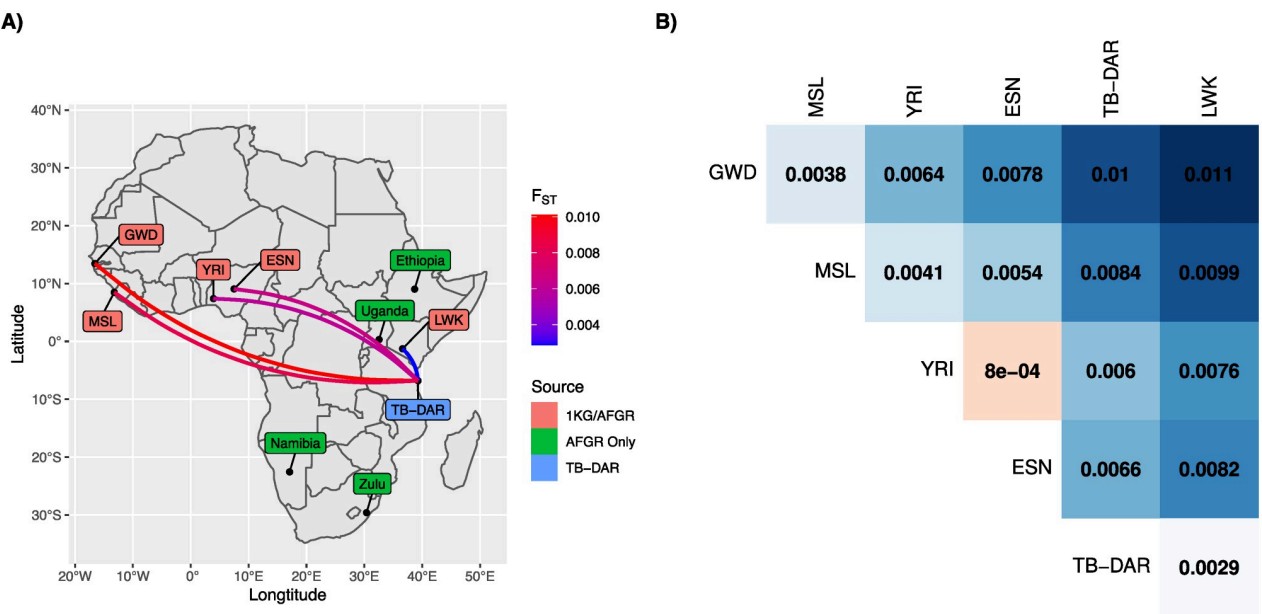

**Fig 2. Genetic differentiation of African populations. A)** Sampling locations of the TB-DAR WGS cohort and populations within the AFGR reference panel, which includes the Sub-Saharan African populations of the 1000 Genomes (1KG) project. Line colors illustrate the degree of differentiation ($F_{ST}$) between TB-DAR and 1KG populations. **B)** Pairwise $F_{ST}$ measures between 1KG populations and TB-DAR. 1000 Genomes Populations: GWD—Gambian in Western Divisions in the Gambia; MSL—Mende in Sierra Leone; YRI—Yoruba in Ibadan, Nigeria; ESN—Esan in Nigeria; LWK—Luhya in Webuye, Kenya. The map was created programmatically in R using the spData package [58], with the base layer based on public domain maps from Natural Earth (https://www.naturalearthdata.com/).

distances in PC space approximately scaled with geographic distances between the sampling locations of populations.

Given the observed genetic differentiation between the TB-DAR and population represented in the AFGR reference panel, we next evaluated the baseline imputation performance of the TB-DAR cohort compared to Sub-Saharan African 1KG populations that are represented in the reference panel. S3 Fig illustrates that compared to all 1KG populations, lower number of variant sites were successfully imputed (INFO > 0.8) in the TB-DAR cohort across all minor allele frequency (MAF) thresholds. For example, for autosomal variants with a MAF of approximately 0.05, 95.8% of variants were successfully imputed in the TB-DAR cohort compared to a mean of 97.7% across the Sub-Saharan African 1KG populations. The difference was more pronounced for the X chromosome, with 89.4% of variants successfully imputed in the TB-DAR cohort compared to a mean of 94.3% across the Sub-Saharan African 1KG populations.

These results quantify the genetic diversity of populations within Africa, and illustrate the differentiation of the TB-DAR cohort from Sub-Saharan African populations of the 1KG project that are represented in the AFGR reference panel. Driven by such differentiation, imputation performance was lower in the TB-DAR cohort when the AFGR reference panel was used. Thus, the need to supplement external reference panels with Tanzanian specific haplotypes and to design population-specific add-ons for the TB-DAR cohort is warranted.

## Selection of add-on tag SNPs and improvements in imputation accuracy

The selection of add-on tag SNPs was conducted under two different settings. Under a coverage-guaranteeing setting (Setting 1), we selected 1869 add-on tags within 337 prioritized TB-

associated regions. In addition, under an efficiency-driven setting (Setting 2), we selected 2503 further add-on tags across the rest of the genome. S4 Fig shows the distribution of all selected tags across chromosomes.

To confirm the validity of our approach, we used the TB-DAR WGS testing set to compare the imputation accuracy of target variants based on three array designs: 1) As a baseline, the H3Africa array without any add-on tags 2) The H3Africa array with random add-on tags 3) The H3Africa array with population-specific add-on tags selected based on the proposed approach. Fig 3 shows the mean imputation quality of target variants that our add-on tags

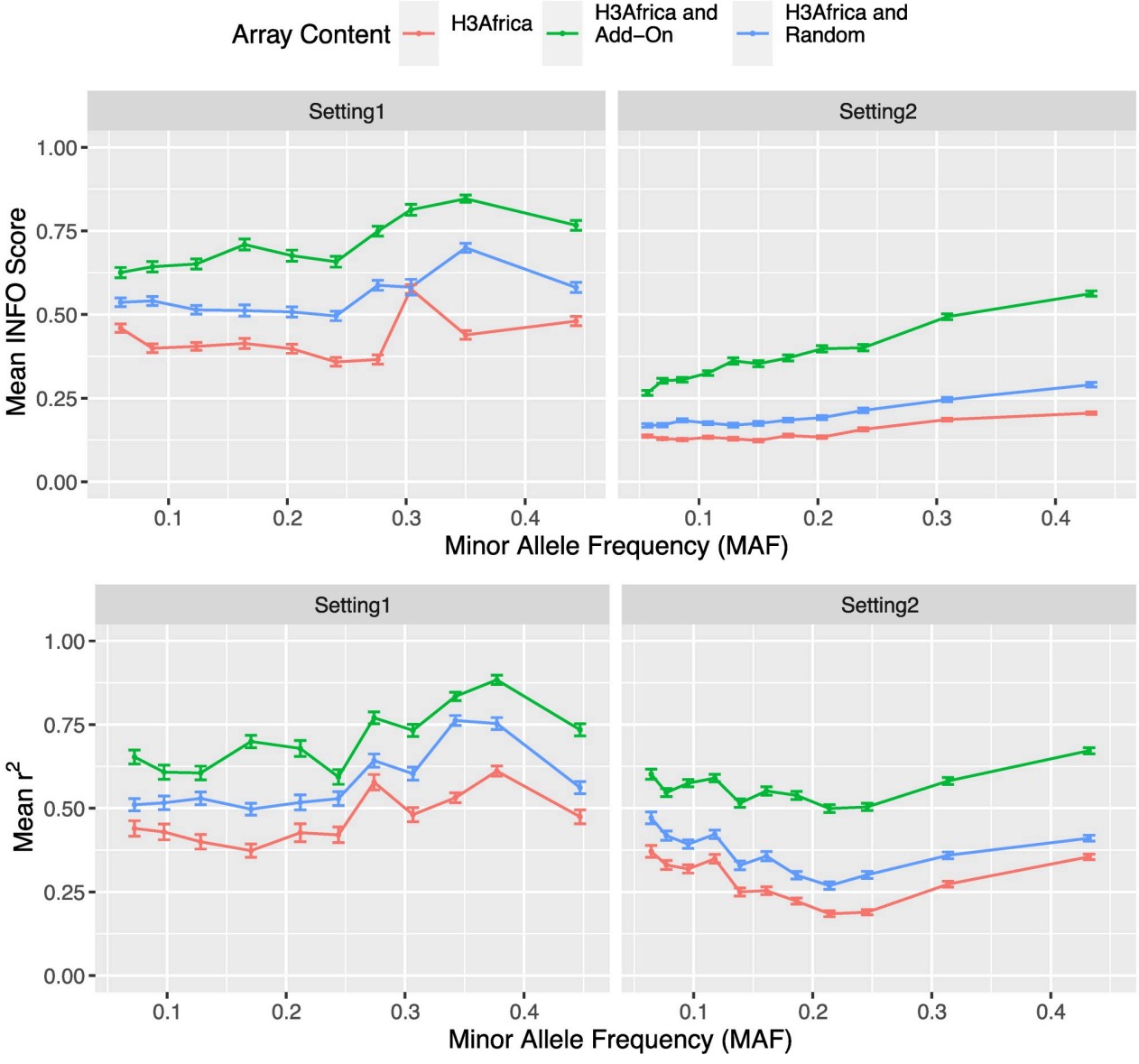

**Fig 3. Improvement in imputation performance subsequent to the addition of add-on tags.** Mean INFO score and $r^2$ (between imputed and sequenced ground truth) of target variants designed to be tagged by add-on tags based on three array designs: 1) H3Africa array without any add-on tags 2) The H3Africa array with random add-on tags 3) The H3Africa array with population-specific add-on tags selected based on the proposed approach. Facet grids illustrate results based on two tag SNP selection settings: coverage-guaranteeing within prioritized regions (Setting 1) and efficiency-driven in all other regions (Setting 2). Error bars represent standard error (SE) of the mean imputation quality within each MAF bin.

were designed to tag across different MAF percentile bins. Under both settings, with the incorporation of add-on tags we observed strong overall improvement in imputation accuracy compared to both the baseline H3Africa array and the H3Africa array with random add-on tags, reflected by the increase in mean INFO score and $r^2$ (correlation with WGS ground truth) across all MAF bins. While the magnitude of increase in mean imputation accuracy was similar for both settings, in general, target variants in prioritized regions were better imputed. This was as intended since, under Setting 1, even relatively well-imputed variants within each region would be tagged by add-on tags to guarantee coverage.

An example region where our approach functioned as expected is shown in Fig 4. Our designed add-on tags lead to improved imputation of target variants compared to both the baseline H3Africa array and the H3Africa array with add-on tags selected by random, reflected by increases in both INFO score and $r^2$. Noticeably, add-on tags were mainly located in

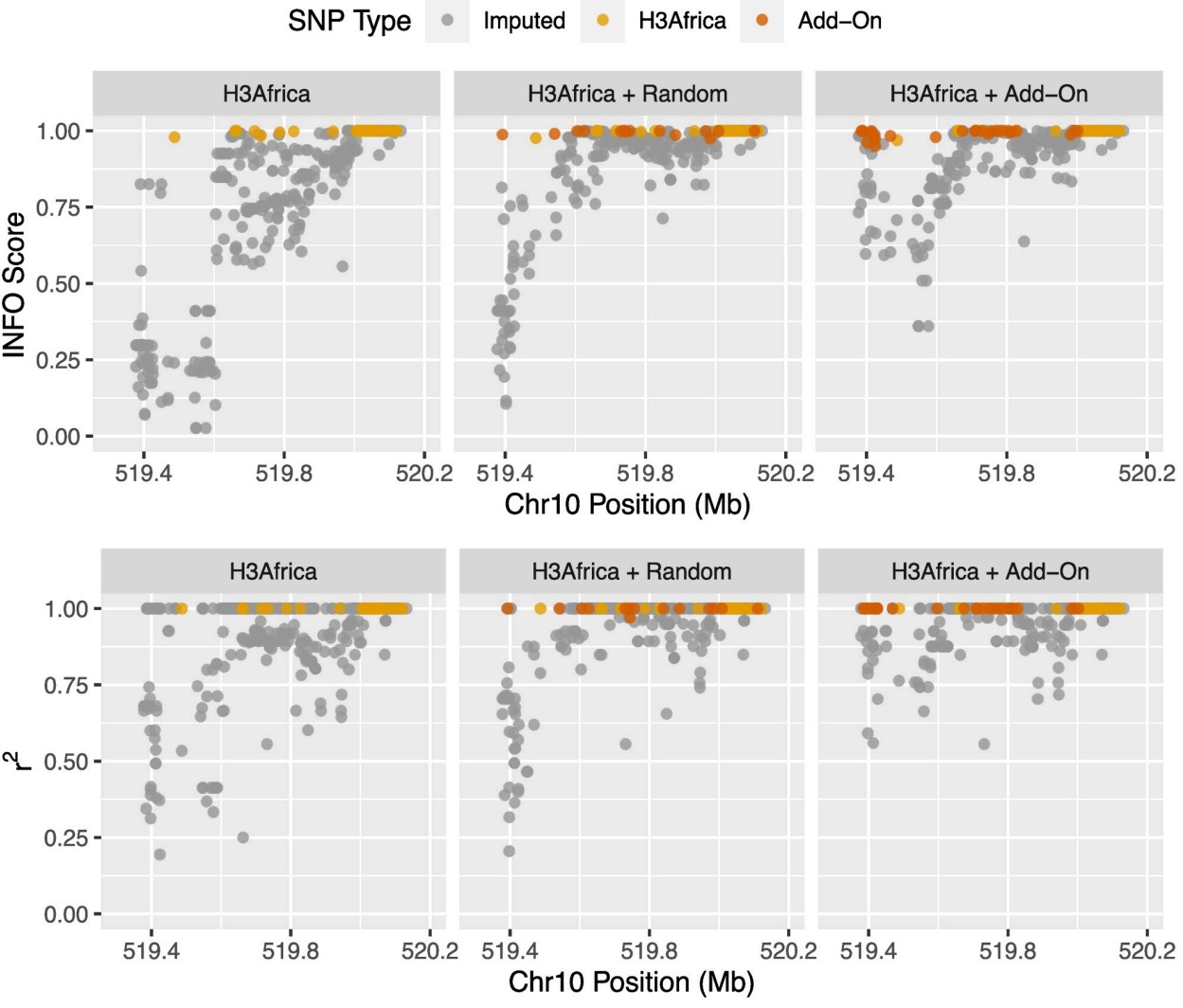

**Fig 4. Improvement in imputation performance in an example region.** Example region on chromosome 10 where the incorporation of add-on tags lead to the increase in imputation performance. Facet grids illustrate imputation performance of the H3Africa array without any add-on tags, with random add-on tags, and with add-on tags selected under the proposed approach. Color of dots represent type of variant (existing H3Africa tags, add-on tags, or any other imputed variants.

proximity to the previously poorly imputed target variants (left side of the region). This indicates, as designed, that only add-on tags that are in relatively strong LD with poorly imputed target variants were selected, as LD generally scales inversely with distance. With the addition of random tags that are approximately uniformly distributed across the entire region, the overall gain in imputation performance was less.

Next, we compared the efficiency of add-on tags selected by the proposed approach to those selected by random and to those selected by the existing Tagger software [23]. We evaluated the number of successfully tagged variants, measured by the number of imputed variants with imputation improvements subsequent to the addition of add-on tags. Our proposed approach explicitly targets variants poorly imputed by either the AFGR reference panel or the internal population specific reference panel. Conversely, Tagger does not rely directly on imputation accuracy to select target variants. Rather, the software infers un-tagged target variants based on existing tags, and subsequently select tags that most efficiently capture these previously un-tagged variants. Table 2 summarises the performance of our proposed approach and Tagger, compared to random selection. Compared to random selection, performance of add-on tags selected by the proposed approach was higher under both Setting 1 and Setting 2. However, compared to Tagger, performance of add-on tags selected by the proposed approach was similar under both Setting 1 and Setting 2. Under Setting 1 (Coverage Guaranteeing), each add-on tag successfully tagged 22.5 variants, while each random and Tagger tag tagged 18.7 and 21.4 variants respectively. Under an INFO score threshold of 0.8 (commonly used in GWAS), this translates to an additional 3196 and 1290 imputed variants being incorporated under our proposed approach compared to random selection and Tagger respectively. Under Setting 2 (Efficiency Driven), each add-on tag successfully tagged 78.3 variants, while each random and Tagger tag tagged 65.8 and 78.1 variants respectively. Under an INFO score threshold of 0.8, this translates to an additional 8260 and 1852 imputed variants being incorporated under the proposed approach compared to random selection and Tagger respectively. For all methods, the number of successfully tagged variants per tag was higher in Setting 2 compared to Setting

**Table 2. Performance of add-on tags, categorized based on settings and methods.** Number of probes (Column 2) indicates the total number of Illumina probes that are required to genotype the add-on tags. The mean probe-ability score (Column 3) estimates the genotyping success rate for the selected add-on probes. The number of successfully tagged imputed variants are measured by either any improvement in INFO score (Column 4), or those exceeding INFO score of 0.8 when previously below (Column 5). Per probe and per tag indicate the number of imputed variants with imputation improvements per add-on tag and add-on probe respectively. Standard error (SE) represents variability of the per tag and per probe metric across different genomic regions. %AFGR and %Tanz indicate the proportion of imputed variants with better imputation accuracy based on the AFGR or internal Tanzanian reference panel respectively.

| | Method | Number of Probes | Mean Probe-ability Score (± SE) | Targets with Improvment in INFO Score | | | | Additional Targets Exceeding INFO Score of 0.8 | | | |
|---|---|---|---|---|---|---|---|---|---|---|---|
| | | | | Per Probe (± SE) | Per Tag (± SE) | % AFGR | % Tanz | Per Probe (± SE) | Per Tag (± SE) | % AFGR | % Tanz |
| **Setting 1 (1869 tags)** | | | | | | | | | | | |
| | Proposed Approach | 2114 | 0.71±0.006 | 19.9±3.2 | 22.5±3.2 | 33.1 | 66.9 | 3.5±0.5 | 4.0±0.5 | 65.8 | 34.2 |
| | Tagger | 2186 | 0.75±0.006 | 18.3±2.9 | 21.4 ±3.0 | 31.8 | 68.2 | 2.8±0.3 | 3.3±0.4 | 62.8 | 37.2 |
| | Random Tags | NA | NA | NA | 18.7±2.4 | 26.8 | 73.2 | NA | 2.3±0.3 | 64.2 | 35.8 |
| **Setting 2 (2503 tags)** | | | | | | | | | | | |
| | Proposed Approach | 2688 | 0.87±0.004 | 72.9±2.7 | 78.3±2.7 | 28.4 | 71.6 | 9.2±0.6 | 9.9±0.6 | 70.7 | 29.3 |
| | Tagger | 2905 | 0.73±0.005 | 67.3±2.5 | 78.1±2.7 | 27.6 | 72.4 | 7.9±0.5 | 9.2±0.5 | 67.1 | 32.9 |
| | Random Tags | NA | NA | NA | 65.8±2.5 | 26.9 | 73.1 | NA | 6.6±0.5 | 75.4 | 24.6 |

1. This is expected since under Setting 1, short haplotypes are tagged to guarantee coverage, thus resulting in the reduced efficiency of each add-on tag.

Another difference between our proposed approach and Tagger is that Tagger does not incorporate information with regards to the number of Illumina probes required for each tag. This is specific to the Illumina platform, where two probes are required to determine the relative intensity of complementary alleles at a locus (A/T or G/C SNPs) while a single probe is sufficient for non-complementary alleles (e.g. A to C or G). Thus, add-on content selected by Tagger can potentially result in a more costly array composition when two add-on tags that have similar tagging efficiency require a different number of probes. This was especially pronounced under Setting 2, where more probes were selected by Tagger compared to the proposed approach with no significant overall gain in the number of successfully tagged variants. Table 2 shows that comparing the proposed method with Tagger, the proposed method had similar number of successfully tagged variants per tag (78.3 and 78.1 respectively), but higher number of successfully tagged variants per Illumina probe (72.9 and 67.3 respectively).

## Mitochondria and Y chromosome haplogroups

Since mitochondrial and Y chromosome haplogroups provide an efficient manner to track human evolutionary history, we targeted haplogroup markers to improve the accuracy of haplogroup calling. The distribution of mitochondrial and Y chromosome haplogroups within the TB-DAR WGS cohort are shown in S5A and S5B Fig respectively. With regards to the mitochondrial DNA, most individuals belonged to the L haplogroup. This was consistent with findings based on the 1000 Genomes project [24], where the L haplogroups were found to be the dominant haplogroups in African populations. For the Y chromosome, a majority of male individuals belonged to the E haplogroups, with a small minority belonging to the B, R, and others. This was also consistent with the 1000 Genomes project [25], where the E haplogroups were found to be dominant in African populations. Also in the Luhya population in neighbouring Kenya a small minority belonged to the B haplogroup [25].

To ensure that our add-on content includes haplogroup markers that complement the existing content on the H3Africa array, we selected 103 and 31 haplogroup marker SNPs as add-ons for the mitochondria and Y chromosome respectively. For the mitochondria, we saw an average improvement in haplogroup calling of 22% compared to the H3Africa array. For the Y chromosome, due to the limited number of add-on tagsand sufficient coverage by the H3Africa array, we did not observe any significant differences in haplogroup calling.

## Discussion

The use of internal population-specific reference panels that supplements external reference panels have been shown to improve genotype imputation, especially in populations that are underrepresented in existing reference panels [18, 19]. Using a Tanzanian cohort as proof-of-concept, our work confirms the utility of internal reference panels. Additionally, we showed that the use of add-on tags jointly with an internal reference panel could further improve the imputation accuracy of common variants.

With regards to application in GWAS, the cost-effectiveness of our approach lies in the balance between sensitivity and power. We envision that our method would be more cost-effective for larger cohorts, as only a small subset of the cohort would need to be whole-genome sequenced under a fixed cost to increase sensitivity for the entire cohort. However, for smaller cohorts, the gain from the increased power through genotyping more individuals using a commercially available array without any add-on content may be more preferable to the sensitivity gained from our approach. Furthermore, array manufactures may also be less willing to offer

add-on customisation for small cohorts. For example, Illumina only offers the possibility of customisation of the H3Africa array for orders with more than 1152 samples.

With the release of larger and more diverse reference panels, the expected gain from the proposed approach is expected to decrease for many populations. The proposed approach can also be time-consuming to implement, due to administrative and logistical factors involved in sequencing, selection of additional variants, and the production of custom arrays. Thus, those interested in applying the approach should first check whether their study population of interest is well-represented in existing array designs and reference panels, and then consider whether the potential sensitivity gained from the proposed approach may be worthwhile. However, until reference panels and corresponding array designs are able to capture almost all of the human genetic diversity, we still believe that our approach fills an important gap for certain underrepresented populations.

Our add-on tag selection procedure did not explicitly target population-specific variants, such as ancestry informative markers [26, 27]. Rather, any common variants in our study population that are poorly imputed under the existing base array content were targeted, and these variants could be either specific to the study population or not. Such a choice was driven by the aim of GWAS, which is to map any variant associated with the trait of interest. Nevertheless, the proposed approach did successfully leverage population specific haplotype structures to improve imputation as designed. This was reflected by the substantial fraction of the tagged imputed variants (66.9% and 71.6%, under Setting 1 and Setting 2 respectively) that were better imputed by the internal Tanzanian reference panel compared to the AFGR reference panel (Table 2).

An add-on tag that most efficiently tags a target variant (in the strongest LD) may not necessarily be the optimal tag or even possible to be assayed on an array, as the genotyping error rate of the probe for the particular tag SNP may be high. In the case of Illumina platforms, the quality of the probe(s) that assay each tag SNP is predicted by a proprietary algorithm that outputs a "probe-ability" score publicly available to researchers. Thus, we were able to rectify such an issue by limiting our selection to add-on tags with probes that have high success rates (Illumina probe-ability score > 0.3), and weighed the trade-off between LD strength and probe quality equally when selecting the optimal add-ons. Nevertheless, a more complex weighting scheme may result in even better performance. Furthermore, while the addition of 5000 probes in this study did not result in the saturation of any bead pools of the array, this may become problematic with a larger number of add-on probes. For such scenarios, our approach would have to be modified to balance the saturation of existing bead pools against the suitability of the probe and the tagging efficiency of the tag.

We introduced two settings for the selection of add-on tags, namely either coverage-guaranteeing (Setting 1) or efficiency-driven (Setting 2). For users of our approach, the number of regions assigned to each setting could be adjusted depending on the study. For example, if there exists strong prior knowledge with regards to genes implicated in or loci associated with the trait of interest, these regions could be assigned to Setting 1. Conversely, for traits with a lack of prior knowledge, a greater proportion of regions could be assigned to Setting 2, such that tag selection would be conducted in a more hypothesis-free manner.

A limitation of our approach is that only common variants in the TB-DAR cohort (MAF > 0.05) were targeted by the selected add-on tags. Such a choice was made due to the limited sample size of our WGS cohort, where for rare variants in the TB-DAR cohort there would be insufficient observations to estimate LD. Nevertheless, the imputation accuracy of low-frequency variants (for example, 0.01 < MAF < 0.05) which are in strong LD with the targeted variants could still increase if tested in a larger testing set. Another limitation of our approach is that the selection of add-on tags within the MHC regions may be suboptimal. Our approach

could be improved by utilizing more accurate variant calling in the MHC regions, for example through the incorporation of alternative contigs of the reference genome [28]. Rather than targeting all common variation in the region, tags could also be selected to tag HLA alleles [29], which could be inferred using HLA allele typing approaches based on the WGS data [30]. Finally, the proposed approach uses a single-marker tagging approach based on pairwise MI to identify add-on content. The use of multi-marker tagging approaches could improve the efficiency of the selected add-on tags [31, 32].

In conclusion, in order to improve imputation accuracy in populations underrepresented in existing reference panels and genotyping array designs, we propose a framework where a subset of a cohort is sequenced and the rest genotyped using an array supplemented with the selected add-on tag SNPs. Using a Tanzanian-based cohort as a proof-of-concept, we demonstrated that under our approach, the WGS data could be leveraged to supplement existing reference panels and to select add-on tags, such that imputation accuracy is improved. Our approach is generalizable to any other population to improve genotype imputation, and thus provides a cost-effective solution to increase the power of GWAS in a diverse range of underrepresented populations and to further our understanding of human genetic diversity.

## Materials and methods

### Ethics statement

Ethical approval for the TB-DAR cohort has been obtained from the Ethikkomission Nordwest- und Zentralschweiz (EKNZ UBE-15/42), the Ifakara Health Institute—Institutional Review Board board (IHI/IRB/EXT/No: 24–2020) and the National Institute for Medical Research in Tanzania—Medical Research Coordinating Committee (NIMR/HQ/R.8c/Vol.I/ 1622). A written informed consent has been obtained from every patient who has been recruited into the TB-DAR cohort. This consent includes the use of the patient's blood for human genomic analyses.

### Study description

This study was conducted based on a cohort of adult pulmonary tuberculosis (TB) patients from Dar es Salaam, Tanzania (TB-DAR). Participants were recruited at the Temeke Regional Hospital in Dar es Salaam. 128 patients were randomly selected from the cohort for WGS, and 116 samples which passed sequencing quality control were retained. Ethnic information of patients are based on self-reported information.

### Whole genome sequencing and quality control

WGS was performed at the Health2030 Genome Center in Geneva on the Illumina NovaSeq 6000 instrument (Illumina Inc, San Diego CA, USA), starting from 1 μg of whole blood genomic DNA and using Illumina TruSeq DNA PCR-Free reagents for library preparation and the 150nt paired-end sequencing configuration. Average coverage was above 30× for 75 samples, between 10× and 30× for 40 samples, and approximately 8× for a single sample.

Sequencing reads were aligned to the GRCh38 (GCA_000001405.15) reference genome using bwa [33] (Version 0.7.17), and duplicates marked using Picard (Version 2.8.14, http://broadinstitute.github.io/picard/). Following the GATK best practices (Germline short variant discovery) [34], Base Quality Score Re-calibration (BQSR) was applied using the GATK package [35] (Version 4.0.9.0). Variants were called individually per sample and then jointly. A Variant Quality Score Re-calibration (VQSR) based filter was then applied, with a truth

sensitivity threshold of 99.7 and an excess heterozygosity threshold of 54.69. Samples with a high genotype missingness rate ($> 0.5$) were excluded.

To ensure that coordinates of the TB-DAR WGS data matched the GRCh37 based AFGR reference panel, a liftover was applied using Picard LiftoverVcf with the UCSC chain file (hg38ToHg19). Only variants that were successfully lifted over to the same chromosome were retained. Within the X and Y chromosomes, variants within the pseudoautosomal regions [36, 37] were excluded.

## Comparison of external reference panels

To compared the performance of external reference panesl, the entire WGS data was masked such that only sites present on the H3Africa array were retained. We measure imputation performance based on three metrics: the imputation quality (INFO score reported by AFGR/HRC and $r^2$ reported by CAAPA), correlation with ground truth measured by squared pearson correlation coefficient ($r^2$), and the fraction of variants observed in the WGS data that were successfully imputed (*INFO* or $r^2 > 0.8$).

## Fixation index and genetic principal components

To conduct principal component analysis (PCA), only autosomal single-nucleotide variants that were genotyped in both 1000 Genomes and TB-DAR WGS cohorts were included. Long-range LD regions were identified separately across all super-populations and across only African populations using the *snp_autoSVD* function of the bigsnpr package [38] in R, and variants within long-range LD regions were excluded. Using PLINK (Version 1.9) [39], LD pruning [40] (`plink --indep-pairwise 1000 50 0.05`) was applied and principal components were derived based on the merged cohorts (TB-DAR and all 1000 Genomes super-populations or TB-DAR and all 1000 Genomes African populations).

To measure differentiation between the TB-DAR WGS cohort and various 1000 Genomes African populations, pairwise fixation index ($F_{ST}$) was calculated using the Hudson estimate implemented in the EIGENSOFT software package (version 8.0.0). Within the TB-DAR WGS cohort, relatedness between individuals was calculated using KING [41], and a random individual in a pair of first degree relatives was excluded. Within the 1KG African populations, relatedness information was obtained from the 1KG project, and individuals labelled as the child, sibling, or grandparent of families or trios were excluded (281 individuals excluded). Only single-nucelotide variants on the autosomes that were genotyped and common (MAF $>$ 0.05) in the merged cohort (TB-DAR and all 1000 Genomes African populations) were included.

## Selection of add-on tag SNPs

Our approach to select add-on tags can be divided into three main steps. In step 1, genotype imputation was performed. Poorly imputed variants were identified, and act as candidate target variants which our add-on tags would be designed to tag. In step 2, the optimal add-on tags were selected based on the population-specific LD structure and allele frequencies of the study cohort. In step 3, we evaluated the improvement in imputation performance when the selected add-on tags were incorporated onto the base H3Africa array.

**Step 1: Genotype imputation and identification of candidate target variants.** The TB-DAR WGS cohort was divided into a training set (3/4 of the data) and a testing set (1/4 of the data).

To achieve optimal imputation accuracy, two reference panels (both internal and external) were used to capture haplotype structures present in both the Tanzanian population and in

other African populations. A internal Tanzanian reference panel based on the TB-DAR WGS training set samples was constructed using Minimac3 [42]. The African Genome Resources (AFGR) reference panel (https://www.apcdr.org/) hosted on the Sanger imputation service (https://imputation.sanger.ac.uk/) [43] was also utilized, where EAGLE2 [44] was used for phasing and the positional Burrows-Wheeler transform (PBWT) [45] was used for imputation. While Minimac3 and EAGLE2 output imputation accuracy metrics that are not strictly comparable ($r^2$ and INFO score), the two metrics have been shown to be highly correlated [46]. Thus, for each site the genotype call was derived based on a direct comparison of imputation accuracy between the two reference panels. The higher score was stored and herein referred to as INFO score.

To identify poorly imputed variants expected under the H3Africa array content (Version 2, https://chipinfo.h3abionet.org/help), the TB-DAR WGS testing set was masked such that only sites present on the H3Africa array were retained. The masked data was imputed using both reference panels, and for each variant, imputation was based on the reference panel that yielded a better imputation score. Candidate target variants were designated as variants that are poorly imputed (INFO < 0.8) but common (MAF > 0.05) in the TB-DAR WGS cohort.

**Step 2: Add-on tag SNP selection.**    For each region, the set of candidate target variants ($S_1$) was defined as variants that are poorly imputed but common in the TB-DAR cohort (MAF > 0.05). The set of candidate add-on tag SNPs ($S_2$) was defined as sequenced common single-nucleotide variants in the TB-DAR cohort, part of the AFGR Reference Panel or the TB-DAR reference panel, with genotype missingness below 0.5, and available as Illumina Infinium probes (probe-ability score > 0.3). The set of existing tags ($S_3$) was initialized as tags that are part of the H3Africa array.

LD information between variants were calculated based on TB-DAR WGS training set. We utilized mutual information (MI) as a LD metric (See S1 Appendix), consistent with the choice of a previous array design study for the Japanese population [47].

To select the optimal set of add-on tags, we followed the framework of a forward-selection based algorithm [47]. In summary, the algorithm select tags that are in the strongest LD with the highest number of candidate target variants not captured by existing tags.

S6 Fig illustrates an example of a single iteration of the add-on tag selection algorithm.

For a single iteration of the add-on tag SNP selection algorithm:

1. For a candidate target variant ($j$), the existing tag that is in strongest LD with it was identified. The MI score of the target variant ($s_j$) was defined as:

$$s_j = \max_{i \in S_3} I_{ij}$$

where $I_{ij}$ denotes the MI between variant $i$ and variant $j$. In S6(A) Fig, SNP4 and SNP5 are the target variants, and SNP3 was identified as an existing tag that is in strongest LD with both target variants.

2. For each pair of candidate add-on tag ($k$) and candidate target variant ($j$), the add-on tag's efficiency was defined as the expected change in MI ($\delta_{jk}$) resulting from the incorporation of the add-on tag:

$$\delta_{jk} = I_{jk} - s_j$$

In S6(B) Fig, SNP1, SNP4, and SNP5 are the candidate tags. SNP4 and SNP5 are the target variants, and the change in MI against both variants ($\delta_{jk}$) were calculated for each candidate tag.

3. The efficiency of a candidate add-on tag ($e_k$) against all candidate target variants was defined based on the sum of the changes in MI:

$$e_k = \frac{\sum_{j \in S_1} \max(0, \delta_{jk})}{N_k}$$

where $N_k$ denotes the number of probes required for the $k^{th}$ candidate add-on tag (This is specific to the Illumina platform, where 2 probes are required for for A/T or C/G SNPs due to complementarity and 1 for all others). In S6(B) Fig, SNP5 is also the most efficient tag, with $e_k = 0.9$ because it requires only a single probe and increases MI against SNP4 and SNP5 by 0.4 and 0.5 respectively.

4. The optimal add-on tag ($k^*$) was identified based on the overall rank of its efficiency and probe-ability scores:

$$k^* = \operatorname*{argmin}_{k \in S_2} r_{e_k} + r_{p_k}$$

where $r_{e_k}$ and $r_{p_k}$ denotes the ranking of the efficiency score and probe-ability score respectively for the candidate add-on tag $k$. In S6(C) Fig, SNP5 is the most optimal tag, since it achieves the lowest overall rank of 3 given a rank of 2 based on probe-ability (S6(D) Fig) and rank of 1 based on efficiency (S6(B) Fig).

5. $k^*$ was added to the set of existing tags ($S_3$), and the above steps were repeated. This is illustrated in S6(E) Fig. The selection procedure was stopped when there are no candidate add-on tags remaining ($S_2$ becomes empty), or when the stopping criteria were met.

**Step 2: Region definitions and stopping criteria.** To ensure the efficiency of add-on tag SNP selection but simultaneously guarantee sufficient coverage in prioritized regions, a two-step procedure for tag SNP selection procedure with unique region definitions and stopping criteria was established.

Under Setting 1, regions spanning 5000 base pairs upstream and downstream of genes or variants associated with TB outcomes (reported by GWAS catalog [48], Open Targets [49], and other GWAS studies [50–52]) were considered. The killer cell immunoglobulin-like receptor (KIR) and human leukocyte antigen (HLA) gene regions were also considered. A region was subject to add-on tag SNP selection if they contained a substantial number of poorly imputed common variants (MAF > 0.05 in the TB-DAR cohort), defined as more than 20% of variants with INFO < 0.8. Regions were also subjected to add-on tag selection if it contained an uneven spatial distribution of well imputed variants, defined as the spread of poorly imputed variants (INFO < 0.8) being more than 1.25 times the spread of well-imputed variants (INFO ≥ 0.8). For example, the spread of well-imputed variants is defined as $\frac{\sigma_{INFO > 0.8}}{max(\sigma_{INFO > 0}, \sigma_{WGS})}$, where $\sigma$ represents the standard deviation of the positions of variants under each criterion (imputed common variants with INFO > 0.8, all imputed common variants, and all sequenced common variants). To guarantee sufficient coverage, iterations of the forward-selection algorithm were run for each region independently until less than 0.5% of candidate target variants within the region showed $\delta_k$ improvements. The process was then repeated for each of the prioritized regions.

Under Setting 2, the selection of add-on tag variants was expanded to any region across the genome that contained poorly imputed common variants (MAF > 0.05 in the TB-DAR cohort). The regions were defined as either a haplotype block (plink --blocks) [39, 53] or a region spanning 5000 base pairs upstream and downstream a candidate target variant,

whichever larger. To maximize the selected add-on tags' tagging efficiencies, a single iteration of the algorithm was run concurrently across all regions. The tag that scored the best across all regions was incorporated. The process was then repeated until the total number of budgeted add-on probes (N = 5000) has been exhausted.

**Step 3: Evaluation of imputation accuracy.** The TB-DAR WGS testing set was utilized to measure improvements in imputation performance enabled by the add-on tags. For all target variants tagged by at least one add-on tag, imputation quality (INFO score) derived from the base H3Africa array was compared against imputation quality derived from the H3Africa array with the addition of add-on tags. In addition, to measure the accuracy of the imputed genotypes, squared Pearson correlation coefficients ($r^2$) were calculated between the imputed genotype dosages (0, 1 or 2) and the ground truth dosages based on the WGS data.

## Selection of random add-on tag SNPs

Matching number of add-on tags were selected by random within each region under both Setting 1 and Setting 2. The criteria for an SNP to be considered as a candidate add-on tag was identical to the proposed approach, except that the candidate tag does not necessarily have to be available as a high-quality Illumina probe (probe-abiliity > 0.3). Add-on tags which were selected by the proposed approach were excluded. In a small number of regions under Setting 1, the number of remaining candidates were not sufficient to achieve a matching number. To avoid bias, a subset of add-on tags that were selected by the proposed approach were also selected by random to achieve a matching number.

## Selection of add-on tag SNPs using tagger

The performance of our proposed approach was compared to Tagger [23], which is a publicly available software for tag SNP selection. Tagger was applied to each region under both Setting 1 and Setting 2 independently using the pairwise mode, and a matching number of add-on tags were selected based on the ranking of tags provided by Tagger. The criteria for an SNP to be considered as a candidate add-on tag was identical to our proposed approach. Since Tagger directly infers the set of un-tagged target variants based on existing array content rather than imputation performance, all variants common in the TB-DAR cohort (MAF > 0.05) were included as input for candidate target variants. In a small number of regions under Setting 1, the number of candidate tags proposed by Tagger was not sufficient to achieve a matching number. To avoid bias, a subset of add-on tags that were selected by the proposed approach were selected by random to achieve a matching number.

## Y chromosome and mitochondrial Haplogroups

The haplogroups of TB-DAR participants were called using HaploGrep2 [54] and yhaplo [55] for the mitochondria and the Y chromosome respectively. The Phylotree mitochondrial [56] and Y chromosome [57] phylogeny databases were used to identify marker SNPs. Marker SNPs for each main haplogroup that any TB-DAR participant was part of were included as add-on SNPs, if not already existing on the H3Africa array. In addition, we added makers SNPs 2 branch points below the main haplogroup that any TB-DAR participant was part of.

## Supporting information

**S1 Table. Self-reported ethnic groups of study participants.**
(XLSX)

**S1 Fig. Scenarios under which add-on tags could improve genotype imputation.** The array population represents the population that the existing genotyping array is designed for. The study population represents the population that one would like to genotype and for which the add-on tags are designed for. **A)** A target variant with a minor allele that is rare in the array population but common in the study population. **B)** An existing tag with a minor allele that is common in the array population but rare in the study population. **C)** The haplotype structure is different between the array population and the study population.
(TIF)

**S2 Fig. Genetic principal components (PCs) based on variants sequenced in both the TB-DAR WGS and 1000 Genomes cohort.** Percent of variance explained by each PC are indicated in brackets. **A)** All 1000 Genomes populations, grouped according to super-populations. **B)** 1000 Genomes African populations.
(TIF)

**S3 Fig. Imputation performance of Sub-Saharan African populations of the 1000 Genomes project and the TB-DAR WGS cohort based on the AFGR reference panel. A)** Fraction of autosomal variant sites successfully imputed (INFO > 0.8) **B)** Fraction of X-chromosome variant sites successfully imputed (INFO > 0.8).
(TIF)

**S4 Fig. Number of add-on tags SNPs on each chromosome and the mitochondria. A)** Add-on tags SNPs selected based on Settting 1 and Setting 2. **B)** Existing tags SNPs on the H3Africa array.
(TIF)

**S5 Fig. Haplogroups of participants within the TB-DAR WGS cohort. A)** Mitochondria **B)** Y chromosome (Males Only).
(TIF)

**S6 Fig. Schematic illustrating a single iteration of the add-on tag SNP selection algorithm.** Edges between variants represent the strength of Linkage Disequilibrium (LD), measured by Mutual Information (MI). The optimal candidate tag SNP is selected based on the best overall rank, taking into account the efficiency ($e_k$), the number of probes required ($N_k$), and the quality of the probe (Illumina probe-ability). In subsequent iterations, the newly added tags are incorporated as existing tags, such that the change in MI ($\delta_{jk}$) includes the contribution of add-on tags.
(TIF)

**S1 Appendix. Detailed derivation of pairwise mutual information (MI).**
(PDF)

## Acknowledgments

We thank the study participants of the TB-DAR study for their contribution. We thank K. Harshman, I. Bartha, C. Howald, and D. Lamparter (Health 2030 Genome Center, Geneva, Switzerland) for sequencing support. We thank C. Thorball, K. Popadin, D. Lawless, O. Naret, and F. Hodel for helpful discussion.

## Author Contributions

**Conceptualization:** Zhi Ming Xu, Sina Rüeger, Jacques Fellay.

**Data curation:** Jerry Hella.

**Formal analysis:** Zhi Ming Xu, Michaela Zwyer.

**Funding acquisition:** Daniela Brites, Damien Portevin, Sebastien Gagneux, Jacques Fellay.

**Investigation:** Faima Isihaka, Hosiana Temba, Thomas Maroa, Jerry Hella, Mohamed Sasamalo.

**Methodology:** Zhi Ming Xu, Sina Rüeger, Michaela Zwyer, Jacques Fellay.

**Project administration:** Jerry Hella, Klaus Reither, Damien Portevin, Sebastien Gagneux, Jacques Fellay.

**Resources:** Daniela Brites, Liliana Rutaihwa, Faima Isihaka, Hosiana Temba, Thomas Maroa, Rastard Naftari, Mohamed Sasamalo.

**Software:** Zhi Ming Xu.

**Supervision:** Klaus Reither, Damien Portevin, Sebastien Gagneux, Jacques Fellay.

**Visualization:** Zhi Ming Xu.

**Writing – original draft:** Zhi Ming Xu, Jacques Fellay.

**Writing – review & editing:** Sina Rüeger, Michaela Zwyer, Daniela Brites, Hellen Hiza, Miriam Reinhard, Sonia Borrell, Jerry Hella, Klaus Reither, Damien Portevin, Sebastien Gagneux.

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
