## [Decision Letter · Decision Letter 0]

5 Jul 2021

Dear Dr. Fellay,

Thank you very much for submitting your manuscript "Using population-specific add-on polymorphisms to improve genotype imputation in underrepresented populations" for consideration at PLOS Computational Biology.

As with all papers reviewed by the journal, your manuscript was reviewed by members of the editorial board and by several independent reviewers. The majority of the reviewers concluded that a major revision would be appropriate. Therefore, in light of the reviews (below this email), we would like to invite the resubmission of a significantly-revised version that takes into account the reviewers' comments.

We cannot make any decision about publication until we have seen the revised manuscript and your response to the reviewers' comments. Your revised manuscript is also likely to be sent to reviewers for further evaluation.

Sincerely,

Alexander Schönhuth

Guest Editor

PLOS Computational Biology

Ville Mustonen

Deputy Editor

PLOS Computational Biology

Reviewer's Responses to Questions

**Comments to the Authors:**

Reviewer #1: The sub-optimal performance of genotyping array in populations that were not represented in the dataset used for its design is well-known. To address this challenge the authors have proposed an approach that would use whole-genome sequencing of representative samples from such a cohort (especially for under-represented populations) for the identification of tag SNPs that are valuable for capturing genetic diversity/LD of the population but are absent in the current genotyping array. Using a computationally improved array they further show that the inclusion of these addon tag SNPs could lead to better imputation and thereby increase the genomic coverage of the dataset. So while this study addresses a key genomic problem I have the following major concerns:

a. My first major concern is that while this approach is sound and effective in theory, the implementation is not straightforward and perhaps not even feasible in some scenarios. Firstly, the cost of sequencing ~100-150 high coverage whole genome sequences is still non-trivial, especially for many African and other under-representative populations. The same amount can be spent to genotype more individuals to increase power of the association study. Secondly, it might be extremely difficult to convince the array manufacturers to add on thousands of novel /continent-specific SNPs, especially if the cohort is small. Thirdly, not all SNPs work on all arrays genotyping platforms, so normally during array design, there is a bit of back and forth between what should go in and what is technically possible, this requires the active participation of both the research group and the manufacturer's bioinformatics team and is time and energy-consuming. Some platforms even need to experimentally validate the SNPs before adding them to the array which could further delay this. Thirdly, oftentimes these arrays are saturated in terms of bead pools so adding new bead pools might not be easy.

b. The improvement of imputation by addition of tag SNPs in the array is expected, so what assumes importance in this study is the extent of the improvement and also a demonstration that the specific method/algorithm that the authors use for SNP selection is leading to improvement beyond what would have been achieved by randomly adding common novel SNPs to the array. Also, comparisons to show that the method used has better (or least similar) performance in comparison to currently available approaches for tag SNP selection is critical.

I would recommend that the authors rewrite this paper focusing on their approach for tag SNP selection and comparing its performance to competing methods.

Reviewer #2: Please see the attached comments

Reviewer #3: The manuscript proposes an approach to increase imputation efficiency on SNP array data from populations poorly represented in public databases. The method consists of submitting a subset of the population of interest to WGS, and then use this information to select a more suitable set of population-specific SNPs to increase the quality of imputation. Their method show to be usefull and can be conveniently applied mainly to data from isolated populations. There are some minor issues outlined below that I recommend to be addressed.

1- Throughout the text (for example in the Figure 1 legend) the authors refers to common and rare SNPs. A genetic polimorphism is a phenomenom related to the allele frequencies of a single genetic locus and can not be rare or common when a single population is considered. They are referring to rare and common alleles it would be better to correct it, making the text more precise and clear. In addition, when the alternative allele of a genetic marker is too rare, by definition it could not even be classified as an SNP.

2- The authors present the add-on SNPs count per chromosome obtained from Setting 1 and Setting 2. Why the pattern of the distribution observed in the barplot of add-on SNPs is so different from that observed for the tag SNPs of the H3Africa array? Wouldn't be expected that longer chromosomes have more add-on SNPs than shorter ones? Besides, Fig. S2-A (Setting 1) shows that the count for chromosome 6 is much higher than the counts obtained from all other autosomal chromosomes. Why? It is known that chromosome 6 includes the MHC region, where genotype determination may present read mapping dificulties related to the short reads generated by high-throughput sequencing. Potential confounding factors for the reliability of MHC region are genotyping the extent of sequence level, structural polymorphism, and the choice of reference sequence. Has any extra care been taken regarding the variant calling of this specific region? Could the MHC region be inflating the chromosome 6 count of Setting 1? Could this be a source of bias?

3- Any missingness data cleaning was performed across markers? Or only at individual level? It is recommended to remove markers with high levels of missing data.

4- Regarding the estimation of within population differentiation, the method used by the authors, although creative, seems to be biased and incorrect. For two main reasons:

(1) The two top principal components of tha PCA acounts for only a small fraction of the total variation (less than 2% for data shown in FigS1);

(2) If the researchers have no information about the existence of population substructure this estimate makes no sense. Otherwise they would have to attribute each individual in the corresponding subpopulation and then estimate Weir & Cockerham pairwise Fst; and this procedure would be correct only if the population is subdivided in exactly two subpopulations. If the researchers have no clue about substructuring and want to check it, the proper way consists in estimate original Wright's fixation index Fst = var(p)/[p(1-p)], which will consider populations with any number of subpopulations.

Since the authors have no reason to believe that there is a substructure in any population, I recommend the exclusion of this analysis or repeating the estimate in a more appropriate way.

Reviewer #4: The review is uploaded as an attachment.

**Have the authors made all data and (if applicable) computational code underlying the findings in their manuscript fully available?**

Reviewer #1: Yes

Reviewer #2: **No: **The genotyping and WGS data are not released, so the described framework here cannot be reproduced by an external researcher.

Reviewer #3: Yes

Reviewer #4: Yes

PLOS authors have the option to publish the peer review history of their article (what does this mean?). If published, this will include your full peer review and any attached files.

Reviewer #1: No

Reviewer #2: No

Reviewer #3: No

Reviewer #4: **Yes: **Gizem Taş
---

## [Decision Letter · Decision Letter 1]

21 Oct 2021

Dear Dr. Fellay,

Thank you very much for submitting your manuscript "Using population-specific add-on polymorphisms to improve genotype imputation in underrepresented populations" for consideration at PLOS Computational Biology. As with all papers reviewed by the journal, your manuscript was reviewed by members of the editorial board and by several independent reviewers. The reviewers appreciated the attention to an important topic. Based on the reviews, we are likely to accept this manuscript for publication, providing that you modify the manuscript according to the review recommendations.

In particular, one reviewer (Reviewer 2) remained to raise somewhat more serious concerns. We would appreciate if you addressed these concerns as a final improvement of the paper. Once we see these addressed satisfactorily, we are inclined to accept the manuscript without further re-reviewing, and quickly proceed with accepting the paper. Please address also the comment on the lack of availability of the studied data.

Sincerely,

Alexander Schönhuth

Guest Editor

PLOS Computational Biology

Ville Mustonen

Deputy Editor

PLOS Computational Biology

[LINK]

Dear Authors,

The

Reviewer's Responses to Questions

**Comments to the Authors:**

Reviewer #1: The authors have addressed most of the main concerns.There in one aspect I would like to comment on. As some of the variable such as admin/MTA, logistics, time delays (probably in the order of 3-6 months -including shipping, sequencing, selection of additional variants based on WGS and their addition to array), which are also major consideration have not been accounted for in this study. I would suggest the authors to add a couple of lines in the discussion to clarify this.

Reviewer #2: I’d like to thank the author for addressing my concerns and implementing new analyses based on my previous review. I think the manuscript is now much clearer. I have some additional comments regarding the current revised manuscript and new analyses:

1. Line 45, the author mentioned the strategy to supplement external reference panels with WGS samples from an internal study cohort. This is possible perhaps in the particular scenario with the AFGR panel. However, this is increasingly impossible – given the sample size of external panels such as TOPMED. It is worth to acknowledge this possibility at some point in the manuscript. In fact, what is lacking in the current manuscript is an actual comparison that AFGR, with or without the internal Tanzanian reference, would outperform the default approach of just using the TOPMED reference panel. It would be good to show that using a smaller AFGR actually is an improvement over TOPMED, and then the authors’ heuristic will further improve upon that.

2. The comparison to tagger shows that the benefit appears to be modest, if any (Table 1). Can there by confidence interval placed on these percentages such that a comparison between Tagger vs. Proposed Approach can be more directly evaluated?

3. At times the writing become too colloquial – for example, the use of “array population” can be confusing, esp. with the similar use of a “study population”. Perhaps the authors can directly define them upfront in text if the only use of these terms are only for convenience.

Reviewer #3: The authors made an excellent work in improving the clarity and precision of the terms used in the text and also in addressing the discussion of methodological limitations related to genetic markers contained in the MHC region.

With the exclusion of the procedure that consisted of dividing the sample into two halves based on the top principal component, the population structure analysis is now correct. The authors were also careful in choosing a method that take into account the differences in sample sizes.

In addition, the section including a comparison between the proposed approach and the tagger software has improved the manuscript substantially, especially given the care taken by the authors who have detailed some important aspects regarding Illumina's genotyping platform that could increase the cost of genotyping process (something that the presented method proposes to reduce).

After a careful review of the manuscript, I congratulate the authors for the great improvement of the text. The article is of great value and I recommend its publication as is.

Reviewer #4: The review is uploaded as an attachment.

**Have the authors made all data and (if applicable) computational code underlying the findings in their manuscript fully available?**

Reviewer #1: None

Reviewer #2: **No: **No information to access the genomic data, thus technically the study cannot be replicated using the exact same data.

Reviewer #3: Yes

Reviewer #4: Yes

PLOS authors have the option to publish the peer review history of their article (what does this mean?). If published, this will include your full peer review and any attached files.

Reviewer #1: No

Reviewer #2: No

Reviewer #3: **Yes: **Renan B. Lemes

Reviewer #4: **Yes: **Gizem Taş

Figure Files:

Data Requirements:

Reproducibility:

References:

---

## [Editor Report · Decision Letter 2]

10 Nov 2021

Dear Dr. Fellay,

We are pleased to inform you that your manuscript 'Using population-specific add-on polymorphisms to improve genotype imputation in underrepresented populations' has been provisionally accepted for publication in PLOS Computational Biology.

Best regards,

Alexander Schönhuth

Guest Editor

PLOS Computational Biology

Ville Mustonen

Deputy Editor

PLOS Computational Biology

---

## [Editor Report · Acceptance letter]

11 Jan 2022

PCOMPBIOL-D-21-00653R2 

Using population-specific add-on polymorphisms to improve genotype imputation in underrepresented populations

Dear Dr Fellay,

I am pleased to inform you that your manuscript has been formally accepted for publication in PLOS Computational Biology. Your manuscript is now with our production department and you will be notified of the publication date in due course.

With kind regards,

Orsolya Voros
